# How self-states help: Observing the embodiment of self-states through nonverbal behavior

**Isabelle Engel**[1], **Maja Dshemuchadse**[2], **Caroline Surrey**[1], **Leander Roos**[1], **Philipp Kanske**[1], **Stefan Scherbaum**[1]*

**1** Department of Psychology, TUD Dresden Technical University, Dresden, Germany, **2** Department of Social Sciences, Hochschule Zittau-Görlitz, Görlitz, Germany

* Stefan.Scherbaum@tu-dresden.de

**Data Availability Statement:** Primary data (CSV format) from the ratings are available via the Open Science Framework (osf.io/bmjf9). Video

## Abstract

The concept of self-states is a recurring theme in various psychotherapeutic and counseling methodologies. However, the predominantly unconscious nature of these self-states presents two challenges. Firstly, it renders the process of working with them susceptible to biases and therapeutic suggestions. Secondly, there is skepticism regarding the observability and differentiation of self-states beyond subjective experiences. In this study, we demonstrate the feasibility of eliciting self-states from clients and objectively distinguishing these evoked self-states through the lens of neutral observers. The self-state constellation method, utilized as an embodied approach, facilitated the activation of diverse self-states. External observers then assessed the nonverbal manifestations of affect along three primary dimensions: emotional valence, arousal, and dominance. Our findings indicate that external observers could reliably discern and differentiate individual self-states based on the bodily displayed valence and dominance. However, the ability to distinguish states based on displayed arousal was not evident. Importantly, this distinctiveness of various self-states was not limited to specific individuals but extended across the entire recording sample. Therefore, within the framework of the self-state constellation method, it is evident that individual self-states can be intentionally evoked, and these states can be objectively differentiated beyond the subjective experiences of the client.

## Introduction

While the notion of different self-states is part of many psychotherapeutic and counseling approaches, it remains scientifically mostly unexplored. In the present study we will use the self-state constellation method as an approach to reveal self-states through embodiment not only as an inner, subjective experience but as something that can be directly distinguished via external observation of the body. To better understand this approach, we will start with a brief overview of common therapeutic approaches to self-states. Subsequently, we sketch the connection of these approaches to the notion of embodiment, followed by a more detailed

recordings of the coaching sessions (from which we derived the material rated in this study) are not publicly available due to data protection regulations.

**Funding:** The author(s) received no specific funding for this work.

**Competing interests:** The authors have declared that no competing interests exist.

description of the here investigated self-state constellation method, before finally moving on to the hypotheses of the present study.

## Different approaches to self-states

The concept of different self-states is part of various psychotherapeutic and counseling approaches. Self-states are known as multiple voices in experiential therapy [1], the dialogical self theory [2] or in the assimilation model [3]. In psychoanalytical approaches they are named as multiple codes [4], in social cognitive approaches as self-states [5], in hypnotherapy as ego states [6], in cognitive-behavioral approaches as schema-modes [7], and in systemic approaches as different parts [8] different sides [9] or system elements [10]. Despite wide-spread acknowledgment of the utility of the concept of self-states within the therapeutic and counseling community, their accessibility remains a challenge as they often go unnoticed in the fabric of everyday human behavior. For therapists and counselors aiming to engage with different self-states, unraveling them within the therapeutic process, typically through conversation, becomes essential. To standardize the identification of self-states a growing number of questionnaires have been developed to measure self-states [11], their strength [12] or their degree of integration [13]. All these procedures are based on self-reflection and self-evaluation and are hence limited since self-states are often hidden from awareness, particularly in emotionally distressing situations. These challenges are reflected in the modest amount of research on self-states, highlighting the difficulty in achieving a more objective understanding of this concept (e.g. [14–16]. Consequently, two significant issues emerge concerning the empirical validation of the general concept of self-states, which we aim to address in this study: First, how can methods beyond reflection and conversation be employed to facilitate the assessment of self-states? Second, can self-states be objectively differentiated beyond subjective experiences using a more objective methodology?

## Embodiment and the self-constellation method

To approach these questions, we anchor our investigation in the concept of embodiment, which posits that the mind and body are intricately connected, and internal processes such as memory, reasoning, or emotional experiences are inseparable from bodily movements and perceptions [17–19]. Cognitions and emotions find expression in bodily sensations, facial expressions, gestures, and movements, creating a reciprocal relationship that influences subsequent thoughts and feelings [20–22]. Building on this foundation, our hypothesis is that involving the body in the therapeutic process enhances the retrieval of different self-states. Moreover, we anticipate that this involvement enables the differentiation of self-states through external observation of the body.

The idea that psychotherapy and counseling extend beyond verbal communication to include bodily interventions has given rise to a diverse array of approaches, such as body psychotherapy, which have gained increasing relevance [23]. Furthermore, using spatial positions in the room to facilitate the therapeutic and counseling process, as seen in practices like chair-work [24, 25] has become well-established. One approach involving the body and taking the metaphor of spatial positioning literally [26], is the family constellations intervention [27]. In this group method, clients select representatives for their family members, positioning them in a room to construct a three-dimensional representation of the family dynamic for therapeutic exploration [28]. While rooted in experiential approaches [29, 30] and predominantly practiced in Europe, this method lacks substantial empirical evidence to date [31]. Nonetheless, it has inspired the development of similar techniques in counseling, business coaching, and organizational development, applying the concept of representing system elements in physical

space [32]. Notably, this approach has been extended to work with self-states, allowing individuals to experience the metaphor of inner positions in real space [33]. For instance, practitioners may place paper tags with labels representing different self-states in the room, enabling clients to physically assume the position of each self-state in the constellation, facilitating an embodied exploration. Termed the "self-state constellation method" in this context, it is gaining practical significance, underscoring the need for its scientific exploration. Moreover, it proves particularly fitting for our research questions, as it facilitates the retrieval of self-states through embodiment and provides a means to differentiate self-states via external observation of bodily expressions.

## The present study

In this study, we will employ the self-state constellation method to uncover self-states through embodiment, not solely as an internal, subjective experience but as a phenomenon directly observable through the client's behavior from an external perspective. Our goal is to empirically differentiate the activation of various self-states by assessing the nonverbal affect exhibited, with evaluations conducted by untrained external observers. The study will utilize muted video material from coaching sessions that employed the self-state constellation method (we use the term coaching referring to a type of counseling with non-clinical subjects with everyday life struggles, that is often referred to as life-coaching [34]). Hence the therapist/counselor is called coach and the patient/client is called coachee.

This video material will be rated be rated by the observers for the nonverbally displayed affect along three primary dimensions (valence, arousal, and dominance; i.e. [35]), comparing the different self-states of coachees (the clients in the coaching process) as defined by their labeled positions in real space. We hypothesize that external observers will be able to reliably distinguish between the emotional properties of different self-states within one coachee. Furthermore, we hypothesize that this ability to distinguish between different states will be replicable across different coachees.

## Method

### Ethics statement

The study was performed in accordance with the guidelines of the Declaration of Helsinki and of the German Psychological Society. Ethical approval was not required according to German law since the study did not involve any risk or discomfort for the rating participants. The rated video material was recorded during coaching sessions that were performed within the curriculum of an accredited psychology Bachelor program. All participants in the video recordings and the rating study were informed about the purpose and the procedure of the study and gave digital informed consent before the experiment.

Data protection measures were coordinated with the local data protection office of Technische Universität Dresden. All data were analyzed anonymously.

### Open practices statement

Primary data (CSV format) from the ratings are available via the Open Science Framework (osf.io/bmjf9). We report all data exclusions (if any), and all relevant measures and manipulations in the study. Video recordings of the coaching sessions are not publicly available due to data protection regulations.

## Sample

The final rater sample includes 49 participants (40 females, 8 males, 1 diverse; $M_{age}$ = 21.57 years, $SD_{age}$ = 4.36 years). A post hoc power analysis using G*Power 3.1.9.7 [36] with N = 49, α = 0.05, and three repeated measures (corresponding to the three self-states) yielded an estimated power of 1-β > 0.99 for all three dimensions valence, dominance and arousal respectively. The sample of rating participants had initially consisted of 88 psychology students as we had aimed for a final sample of > 40 participants and had taken into account the expectable loss in online studies. These participants were recruited through an online platform for psychological studies at Technische Universität Dresden between May and July 2021. We excluded 37 individuals who dropped out of the experiment early, one who took four hours instead of the intended 1.5 hours, and one who did not change baseline scores on the rating scales in 229 out of the 360 trials. There is no data for 19 of the 37 drop outs since they left the experiment before any data was logged. Among the remaining drop outs 16 were female, 2 were male, and 1 was diverse ($M_{age}$ = 23.95 years, $SD_{age}$ = 6.52 years). Participants received course credit for their participation.

## Material

**Stimulus material from the intervention.**   The study used video recordings of practice coaching sessions within the curriculum of the Bachelor program in psychology at the Hochschule Zittau/Görlitz. Students led the coaching sessions as coaches and recruited their coachees. This coachee sample for the videos consisted of ten individuals (5 females, 1 male, 4 no gender specification; $M_{age}$ = 23.00 years, $SD_{age}$ = 2.97 years, 6 no age specification) who agreed to the use of the recording of their coaching session for research purposes and met the videoclip selection criteria described below. The initial coachee sample had consisted of 37 individuals. However, 27 (17 females, 4 males, 6 no gender specification; $M_{age}$ = 23.81 years, $SD_{age}$ = 4.01 years, 6 no age specification) of those did not meet the video selection criteria described in the next section. During the intervention, the coach and coachee first determined a topic the coachee wanted to work on. The states were then identified throughout the conversation, relying on self-reflective, conscious processes in this part of the coaching session. In the following coaching process these states within the coachee were marked spatially across the room using prompt cards. Finally, the coachees were guided to the different spatial positions to activate the respective states. This process was repeated two times; in each round the participant was free to activate each state as many times as they wished. Though the intervention manual (see S1 Appendix) aimed to standardize the setting, we refrained from constraining the locations and viewing directions, to stay close to a naturalistic setting and keep the intervention as effective and process-related as possible. The coaching sessions were recorded from three perspectives synchronously using professional observation equipment (Mangold International GmbH). The audio track was separated from the video and discarded for reasons of data protection.

**Video clip selection.**   We aimed to collect 4 ratings for 3 self-states of a coachee and a baseline rating before the start of the session (13 videos per coachee). To keep the experiment within a reasonable timeframe for an online experiment, we decided to use video clips of approximately 15 seconds, which would result in 130 videos with an overall length of approximately 30 minutes. Each video clip was selected according to a set of criteria; for example, the coachee needed to be currently talking as one of their self-states (not as themselves) and not interacting with the coach (see S2 Appendix for a detailed description of the process).

We selected 10 coaching sessions for the best recording quality and angle of sight due to the naturalistic setting (see S2 Appendix). For each of these coaching sessions, we first selected the

**Table 1. Names of the selected self-states for each coachee.**

| Coachee | Self-State 1 | Self-State 2 | Self-State 3 |
|---|---|---|---|
| 1 | Rest-needing | Doubting | Be-Cared-For |
| 2 | Student | Family Person | Responsible |
| 3 | Independent | Relationship-Deepening | Justice Maker |
| 4 | Rational | Anxious/ Intuitive | Free-spirited |
| 5 | Frugal | Adventurous | Creator |
| 6 | Suppressing | Caring | Spiritual |
| 7 | Unleashed | Pragmatic | Academic-Inquisitive |
| 8 | Logician | Rebel | Overseer |
| 9 | Diplomat | Constant | Questioning |
| 10 | Adventure Seeker | Vulnerable | Critic |

self-states that marked the outermost points of the constellation. Out of these states we selected the ones which were represented most frequently during the intervention (indicating their relevance to the coachee) and the ones for which the coachee felt the strongest emotion as rated by the respective coach after the intervention, narrowing down the pre-selection to three states. A list of the selected self-states and their names is shown in Table 1.

**Assessment of the nonverbally displayed affect.** To evaluate coachees' body language concerning their exhibited emotional responses to various self-states, we opted for a widely used and easily applicable tool—the self-assessment manikin [35]. Grounded in a dimensional perspective of affect, this tool characterizes emotions by comparing them along three dimensions [37, 38] and has found extensive application in assessing affective stimuli [35, 39]. The tool assesses three primary affective dimensions—emotional valence, arousal, and dominance/power—each represented pictorially on a Likert scale. Since our aim was primarily, to distinguish different affective states independently of the specific emotion, this assessment was much more efficient, than tools based on a complete set of basic emotions would have been [40]. Furthermore, the used pictorial mode—manikins displaying different non-verbal emotional reactions—is analogous to the task of visually assessing the coachees' body language, and hence should lead to straightforward reactions. We used a 20-point version of the scale [41] along the five manikins per dimension that illustrate possible expressions. This way, raters could tick points right underneath each manikin as well as in-between.

## Procedure

We implemented the study online using the Labvanced platform [42]. Participants could choose the time of participation freely but were asked to find a quiet environment with a stable internet connection for the experiment. After giving their consent to participate in the experiment, the processing of their data, and confidentiality regarding the content of the video clips, participants watched 130 videos. All video clips of one coachee were shown consecutively in randomized order; the order of coachees was also randomized. Following each video clip, participants rated the appearance of the coachee in the clip for the three dimensions (valence, dominance and arousal) via the Self-Assessment Manikins.

## Statistical analysis

We first assessed the inter-rater reliability of our ratings by computing a two-way agreement average-measures intra-class correlation.

Then we used a repeated measures analysis of variance (ANOVA) to test our hypothesis that participants would rate the nonverbal affect of each state of a coachee differently from that of the other states. Conducting this statistical analysis in this setting across all coachees is complicated by the fact that each coachee determines and works with their individual states in the intervention. Hence, this precludes a simple statistical comparison across coachees. Accordingly, we proceeded in two steps. As a first step, we used an ANOVA with the factor *self-state* for each coachee separately.

We set $\alpha \leq \frac{0.05}{10}$ = .005; taking into account the Bonferroni correction for multiple testing (10 ANOVAs per hypothesis/dimension). When the normal distribution of the data was not given, we refrained from using alternative nonparametric tests, because we aimed for a common statistical test across coachees for the following analyses and because ANOVAs are generally robust to violation of the normal distribution assumption [43]. We tested the sphericity of the data using the Mauchly test. In the case of violation of the sphericity assumption, we adjusted the degrees of freedom according to Greenhouse-Geisser.

This ANOVA was performed with the 49 participants' ratings of each coachee's videos for each of the three dimensions. Thus, we obtained effects for each coachee across their individual self-states separately for each dimension.

Lastly, we combined these effects meta-analytically using Cohen's *f* effect size across coachees so that we could estimate an effect of difference between self-states for each dimension.

## Results

### Inter-rater reliability for the affective dimensions

The inter-rater reliability for the assessed affective dimensions valence and dominance indicates moderately acceptable agreement between the participants on the assessment of these dimensions. The inter-rater reliability for the arousal dimension indicates low agreement (see Table 2; [44]).

### Individual differences in the affective dimensions between the self-states of each coachee

An individual analysis of differences between the observed affect of the self-states of each coachee shows medium to large effects for the dimensions valence and dominance and small effects for the dimension arousal (see Table 3; [45]).

### Meta-analytical summary of the differences in the affective dimensions

A meta-analytical summary of the effects across coachees shows significant differences in the observed affective dimensions valence and dominance (see Fig 1).

Across coachees, according to [45], there were large differences between self-states in terms on the valence dimension (significant for 8 out of 10 coachees, $M_f$ = 0.66, $SD_f$ = 0.29) and on

**Table 2. Inter-rater reliability.**

| Dimension | Coachees | | | | | | | | | | |
|---|---|---|---|---|---|---|---|---|---|---|---|
| | 1 | 2 | 3 | 4 | 5 | 6 | 7 | 8 | 9 | 10 | Mean |
| Valence | .701 | .854 | .905 | .590 | .942 | .918 | .908 | .965 | .358 | .968 | .811 |
| Dominance | .695 | .300 | .670 | .112 | .725 | .383 | .929 | .873 | -.143 | .950 | .549 |
| Arousal | -.144 | -.177 | .093 | .476 | -.433 | .481. | .431 | .428 | .092 | .868 | .212 |

**Table 3. Individual differences between the self-states.**

| Coachee | Valence | | | Dominance | | | Arousal | | |
|---|---|---|---|---|---|---|---|---|---|
| | 95% CI | $p$ | $\eta_p^2$ | 95% CI | $p$ | $\eta_p^2$ | 95% CI | $p$ | $\eta_p^2$ |
| 1 | [0.04, 0.28] | < .005* | 0.15 | [0.10, 0.38] | < .005* | 0.25† | [0.00, 0.06] | .66 | 0.009† |
| 2 | [0.10, 0.37] | < .005* | 0.24† | [0.00, 0.16] | .05 | 0.06† | [0.00, 0.05] | .77 | 0.006† |
| 3 | [0.14, 0.28] | < .005* | 0.28 | [0.04, 0.28] | < .005* | 0.15 | [0.00, 0.11] | 0.22 | 0.03† |
| 4 | [0.01, 0.20] | .01 | 0.09† | [0.00, 0.12] | .19 | 0.03† | [0.02, 0.24] | < .005* | 0.12 |
| 5 | [0.22, 0.50] | < .005* | 0.37† | [0.08, 0.34] | < .005* | 0.21†,‡ | [0.00, 0.03] | .90 | 0.002 |
| 6 | [0.19, 0.46] | < .005* | 0.34† | [0.00, 0.18] | .03 | 0.07† | [0.02, 0.25] | < .005* | 0.13 |
| 7 | [0.20, 0.48] | < .005* | 0.35†,‡ | [0.37, 0.62] | < .005* | 0.51‡ | [0.00, 0.18] | .02 | 0.07†,‡ |
| 8 | [0.42, 0.65] | < .005* | 0.55‡ | [0.13, 0.41] | < .005* | 0.28† | [0.00, 0.19] | .02 | 0.08† |
| 9 | [0.00, 0.14] | .12 | 0.05 | [0.00, 0.07] | .56 | 0.01† | [0.00, 0.11] | .22 | 0.03† |
| 10 | [0.46, 0.67] | < .005* | 0.58†,‡ | [0.35, 0.60] | < .005* | 0.49‡ | [0.14, 0.41] | < .005* | 0.28†,‡ |

†: normal distribution not given

‡: sphericity not given, degrees of freedom adjusted according to Greenhouse-Geisser

*significant after Bonferroni correction α = 0.005

the dominance dimension (significant for 6 out of 10 coachees, $M_f = 0.49$, $SD_f = 0.30$). Differences between self-states on the arousal dimension were rare and at most medium-sized (significant in 3 out of 10 coachees, $M_f = 0.25$, $SD_f = 0.18$).

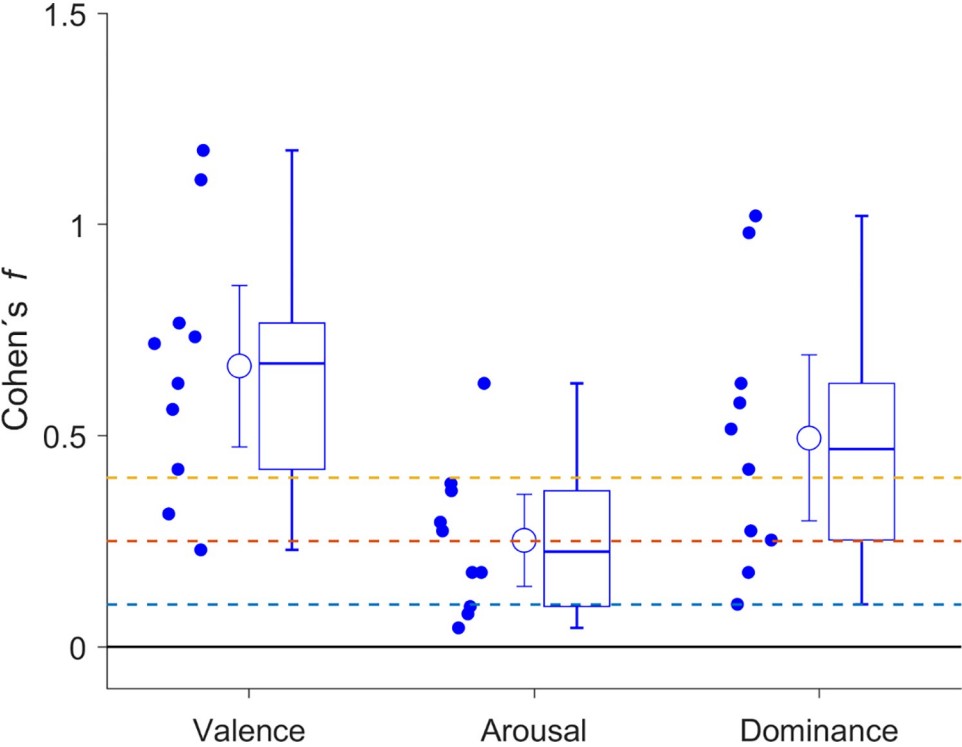

**Fig 1. Meta-analytical summary of differences between self-states.** Effect size $f$ for the three dimensions valence, dominance, and arousal measured with the self-assessment manikins. Each point represents one coachee's effect size, summarized by the mean and 95% CI. The adjacent boxplots mark the median, with boxes marking the 25th and 75th percentiles and error bars marking the extreme edges of data distribution (1.5 of the inner 50% of data). The dashed lines indicate typical levels of effect size for Cohen's $f$ (0.1 small, 0.25 medium, 0.4 large).

## Discussion

This study aimed to provide an objective differentiation of self-states, as utilized in various psychotherapeutic and counseling approaches, thereby opening avenues for the exploration and utilization of self-states. Employing the self-state constellation method, we activated different self-states, and external observers evaluated the nonverbal expressions on three primary dimensions: emotional valence, arousal, and dominance.

Our findings reveal the successful differentiation of coachees' nonverbal emotional reactions by observers, particularly in terms of emotional valence and dominance ratings. This distinctiveness extended not only among the self-states of specific recorded coachees but also across coachees within the entire recording sample. However, ratings on the arousal dimension did not yield reliable differences between the self-states of the coaches.

### The self-constellation method

We successfully achieved our goal of empirically differentiating the activation of various self-states through the observation of displayed nonverbal affect. These activated states were objectively observable, reinforcing the notion that self-state activation involves more than a mere cognitive-reflective process; it incorporates bodily and postural components. This insight suggests that such components can be effectively utilized in subsequent therapeutic interventions. This finding encourages the use of different therapeutic approaches that address the concept of self-states as cited in the introduction. They show that self-states go beyond a catchy metaphor mirroring observable bodily expressions. Following this research path further could strengthen the scientific foundation of self-state methods in therapeutic settings and enlarge their scope of clinical applications. Furthermore, there do exist a wide array of variations in the field of constellation-methods [28], one of them allowing the addition of new spatial positions during the experiential work in real space. This would extend the involvement of the body to the initial phase of self-state identification. For reasons of controllability of the procedure, we did not allow for the addition of new spatial positions during the experiential work in real space.

### Embodiment

From an embodiment standpoint, our results exhibit consistency: activated self-states exert an influence on bodily and postural states, which, in turn, are objectively discernible to outside observers. In alignment with the embodiment approach, this underscores the perspective that cognitions and emotions find expression through observable bodily responses [20–22]. Hence, if the self-state constellation method was able to activate the differing states, one would then expect that these are distinguishable through the observation of non-verbal body expressions. Indeed, our results indicate that the activation of the self-states through entering the labeled-spatial position was distinguishable. This becomes especially relevant when one incorporates the second important assumption of embodiment approaches, namely that the bodily responses feed back into cognitions and emotions. Hence, one could expect that if the bodily states are observable from the outside, they should be strong enough to feed back into the cognitive and emotional part of the therapeutic/coaching process and provide a resource that coachee and coach can harness for their common work on the coachee's problems. Consequently, our results provide empirical evidence for approaches that expand psychotherapy beyond conversation to bodily interventions and stress the importance of more high-quality studies in this area [23].

Beyond its therapeutic implications, our study has the potential to contribute to the broader research landscape on embodiment and the interplay between emotion and cognition. The

results demonstrate that establishing distinct self-states through verbal instructions tied to spatial positions leads to diverse bodily expressions along affective dimensions. These findings could serve as inspiration for ongoing endeavors aimed at integrating three classes of emotion theories: basic emotion theories, constructionist theories, and appraisal theories [46, 47]. The methodology employed in this study offers a valuable approach to investigate emotions within a framework that incorporates various psychological components, such as semantic concepts, appraisals, and bodily expressions. This integration facilitates the observation of intricate emotional patterns.

## Differences between the affective dimensions

Although our participants' ratings consistently differentiated between self-states on the dominance and valence dimensions of the SAM, notable differences on the arousal dimension were observed in only three out of ten coaching sessions. This discrepancy primarily stems from the challenge of reliably identifying arousal, as reflected in low inter-rater reliabilities among observers. While this doesn't pose a critical issue for our primary objective of establishing a general distinctiveness of emotional patterns in different self-states, two potential explanations may account for these observations. First, it is plausible that arousal is inherently challenging to observe in the specific video clips selected. While the psychological literature on emotions unequivocally recognizes arousal as a state manifested in bodily responses [38], a recent study has indicated that across various modalities, arousal is predominantly expressed through the frequency spectrum [48] However, the brevity of the video clips employed in our study, lasting only 15 seconds, might have limited the reliability of extracting this signal information. Consequently, the precision in encoding arousal by our observers may have been compromised. Indeed, this is indicated by a higher overall standard deviation of the arousal ratings (SD = 3.1) compared with the valence (SD = 2.2) and the dominance dimension (SD = 2.8). Second, the situation in the laboratory could have triggered a general increase in arousal that confounded the effects of the self-states. Indeed, the mean of the ratings of the arousal dimension (mean = 11.1) was higher compared with the valence (mean = 9.6) and the dominance dimension (mean = 9.0).

## Generalizability

While the results of our study hold promise, it is crucial to acknowledge that this research marks just an initial stride, and more extensive investigations are necessary to solidify the scientific underpinnings of the diverse therapeutic approaches to self-states. Hence, we address four limitations on the generalizability of our results.

One limitation regarding generalizability is our exclusive use of a coaching setting to examine the observability of self-states, rather than the more established backdrop of psychotherapy. This choice had practical considerations; firstly, the researchers had access to the coaching setting and could control the coaching process within this setting to produce the best possible material for the ratings. Secondly, a coaching setting is less negatively influenced and sensitive to scientific investigation compared to the highly intimate setting, especially the one of psychotherapy. Regarding the validity of our findings, conducting the study in a coaching setting enhances the interpretability for two reasons. First, even in the comparatively less emotional coaching environment, the self-state constellation method effectively activated self-states, inducing bodily states distinguishable from an external perspective. It is reasonable to expect that in therapeutic contexts, self-states could be even more pronounced and distinct. Second, in psychotherapy (especially), certain psychopathological conditions might entail a dissonance between bodily expression and emotional experience, potentially masking the phenomena

observed in this study. Nevertheless, future research should focus on validating and extending our findings to diverse therapeutic settings.

A second limitation is that our study focused on just one among many self-state methods available. We specifically chose a method where self-states are not predefined or categorized, in contrast to approaches relying on, for example, the theoretical differentiation of transactional analysis [49] or the identification of schema modes via clinical observation [50] (note, that the number of differentiated schema modes has also been growing from 18 to 80 in the recent years [51, 52]). This choice aligns with systemic-constructivist approaches, allowing the coachee to determine which self-states are relevant to the coaching process [53]. However, we recommend adapting the self-state constellation method to more structured approaches. This adaptation could offer more objective research methods beyond theory and clinical observation, providing a foundation for studying various self-state approaches.

A third limitation concerns the way how our observers rated the affective content of the observed self-state. The self-assessment manikin is a well-established tool to rate affect. However, in its simplicity, it captures only three basic dimensions of affect. A more nuanced and/or briader approach, eg. the rating of basic emotions, might yield more details about differences in observable affect of self-states. One might even consider to ask for affective idiosyncratic descriptions of the observed self-states to get a broader picture of how the self-states modulate the outer appearance and what is inter-subjectively observable.

Fourth and finally, it is essential to note that the current study comprises a limited number of coaching sessions within a specific study context, and the robustness of our results needs validation. Consequently, future research endeavors should replicate our findings in diverse contexts, employing varied coachee and rater samples to enhance the generalizability of our conclusions. Additionally, to further enrich the present research, alternative, potentially more nuanced scales could be employed to evaluate self-states. Furthermore, investigating the temporal stability of the observed self-states would be a valuable extension of the study.

## Conclusion

While numerous therapeutic and counseling approaches presuppose and engage with self-states, both theoretical and empirical research on this concept remains notably limited and challenging to comprehend scientifically [16]. Here, we showed that outside observers can indeed identify and reliably distinguish different self-states by their nonverbal affect within a coaching setting based on the self-state constellation method. This humble, yet empirically based evidence suggests, firstly, that self-states are a scientifically accessible concept and, secondly, that they could be constructs that are effective in the therapeutic and counseling process. Moreover, our findings endorse therapeutic approaches that expand beyond traditional conversation to incorporate experiential methods involving bodily states, particularly self-states constellations. Lastly, our study advocates for the convergence of diverse therapeutic and counseling approaches around the concept of self-states. Despite distinct differences, these research fields could benefit from amalgamating the accumulated insights into a unified concept, for which we propose the term "self-state" here. Building upon this foundation, future research can not only focus on therapeutic and counseling efficacy but also delve into understanding the underlying processes and mechanisms that drive these effects.

## Supporting information

**S1 Appendix. Manual of the self-constellation method.**
(PDF)

**S2 Appendix. Video clip selection.**
(PDF)

## Acknowledgments

### Transparency statement

The material reported in this manuscript has not been published and is not under consideration for publication in other international/English-language journals. A German short version about a summary of the results has been published in the journal "Hypnose—Zeitschrift für Hypnose und Hypnotherapie" as part of a conference contribution that was not peer-reviewed with the title „Innere Vielfalt sichtbar machen: Unterscheidbarkeit von Inneren Anteilen in Aufstellungen durch naive Beobachtende"[54]. This short version makes the findings of this study available to the national–non-English speaking–community.

## Author Contributions

**Conceptualization:** Isabelle Engel, Maja Dshemuchadse, Leander Roos, Stefan Scherbaum.

**Formal analysis:** Isabelle Engel, Stefan Scherbaum.

**Investigation:** Maja Dshemuchadse, Leander Roos.

**Methodology:** Leander Roos.

**Project administration:** Maja Dshemuchadse.

**Software:** Isabelle Engel.

**Supervision:** Caroline Surrey, Stefan Scherbaum.

**Writing – original draft:** Isabelle Engel, Maja Dshemuchadse, Stefan Scherbaum.

**Writing – review & editing:** Maja Dshemuchadse, Caroline Surrey, Leander Roos, Philipp Kanske, Stefan Scherbaum.

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
