## [Decision Letter · Decision Letter 0]

12 Dec 2023

PONE-D-23-17809How Self-States help: Observing the Embodiment of Self-States through Nonverbal BehaviorPLOS ONE

Dear Dr. Scherbaum,

Thank you for submitting your manuscript to PLOS ONE. After careful consideration, we feel that it has merit but does not fully meet PLOS ONE’s publication criteria as it currently stands. Therefore, we invite you to submit a revised version of the manuscript that addresses the points raised during the review process.

The reviewers highlighted major revisions that should be made in order to evaluate the work and I agree with what has been highlighted. Furthermore, the text has some inaccuracies and it is recommended to use a proofreading service.

We look forward to receiving your revised manuscript.

Kind regards,

Giulia Ballarotto

Academic Editor

PLOS ONE

Journal Requirements:

3. We noted in your submission details that a portion of your manuscript may have been presented or published elsewhere. [A german short version about parts of the results is published in the german journal “Hypnose” as part of a conference contribution with the title „Innere Vielfalt sichtbar machen: Unterscheidbarkeit von Inneren Anteilen in Aufstellungen durch naive Beobachtende“. This short version reports extracts of the analyses as reported in full here and makes the results available to the national – non-english speaking – community.] Please clarify whether this [conference proceeding or publication] was peer-reviewed and formally published. If this work was previously peer-reviewed and published, in the cover letter please provide the reason that this work does not constitute dual publication and should be included in the current manuscript.

4. We note that you have referenced (Bernstein, D. P., & van den Broek, E. et. al(2009)) which has currently not yet been accepted for publication. Please remove this from your References and amend this to state in the body of your manuscript: (Bernstein, D. P., & van den Broek et al. [Unpublished]”) as detailed online in our guide for authors

Reviewers' comments:

Reviewer's Responses to Questions

**Comments to the Author**

1. Is the manuscript technically sound, and do the data support the conclusions?

Reviewer #1: Yes

Reviewer #2: Yes

2. Has the statistical analysis been performed appropriately and rigorously? 

Reviewer #1: I Don't Know

Reviewer #2: Yes

3. Have the authors made all data underlying the findings in their manuscript fully available?

Reviewer #1: Yes

Reviewer #2: Yes

4. Is the manuscript presented in an intelligible fashion and written in standard English?

Reviewer #1: Yes

Reviewer #2: Yes

5. Review Comments to the Author

Reviewer #1: Abstract has not standard way of scribe. In abstract is not suitable to use questions. Also is not right to use abstract as a shortened version of paper contain. I propose complete revision of abstract.

The section Introduction is very long but interest. Authors in this section inform reader about research approaches to the different self-states psychotherapeutic and counselling. Becouse the section Introduction is very long, I propose divided this section the section Introduction and section Related work. Section Introduction should be short and concise (contain short information about current state of research problematic and short information what each of the following sections is about). The methods was clar and replicable. I have reservations only to the size of sample. The size 37 persons is very small sample for inferring of research conclusions. As self-assessment authors use Self-Assessment Manikin - that is very suitable method but missing psychologist's opinion and point of view.

Reviewer #2: The paper focuses the psychotherapeutic relevance of the self-states, a subject of great clinical relevance.

I have some comments

Ethics statement

I’ve a concern about that authors have not required the ethical approval. Which kind of data protection were adopted?

Sample

Have been studies the profile of the students that dropped out early?

Self-Asssessment manikis

A description of the dimensions assessed should be included.

Why the authors have not studied the dimension of affect?

Discussion

I think that this part is the most weak. It should be discussed the relevance and the clinical application, not only in psychotherapy but also in other fields (emotion recognition,...).

It also the diferent dimensions are not discussed and only in the limitation part are included a discussion about arousal dimension.

The self-constellation method

Part of this paragraph should be included in the methodology section.

Limitations

The authors should explain clearly that for make a generalization more studies should be done... even to reply their results, not only in psychotherapy settings. I think that it could be introduced also in the discussion, not in the limitation section.

The comment of arousal dimension is not a limitation, is subject of discussion (see above).

6. PLOS authors have the option to publish the peer review history of their article (what does this mean?). If published, this will include your full peer review and any attached files.

Reviewer #1: No

Reviewer #2: No

---

## [Author Response · Author response to Decision Letter 0]

6 Feb 2024

Editor comments

The reviewers highlighted major revisions that should be made in order to evaluate the work and I agree with what has been highlighted. Furthermore, the text has some inaccuracies and it is recommended to use a proofreading service.

Response: Thank you for pointing out the inaccuracies in the previous version of our manuscript. We have rewritten the mentioned parts and also took great care to increase the accessibility and language quality of the entire manuscript. It has now also been checked by a native speaker.

We noted in your submission details that a portion of your manuscript may have been presented or published elsewhere. [A german short version about parts of the results is published in the german journal “Hypnose” as part of a conference contribution with the title „Innere Vielfalt sichtbar machen: Unterscheidbarkeit von Inneren Anteilen in Aufstellungen durch naive Beobachtende“. This short version reports extracts of the analyses as reported in full here and makes the results available to the national – non-english speaking – community.] Please clarify whether this [conference proceeding or publication] was peer-reviewed and formally published. If this work was previously peer-reviewed and published, in the cover letter please provide the reason that this work does not constitute dual publication and should be included in the current manuscript.

Response: This earlier report of our results does not constitute dual publication for the following reasons:

- First, the report was based on a poster presentation at a conference that was not peer reviewed. It was published in the German journal Hypnose together with the other conference contributions and is hence merely part of the conference proceedings.

- Second, the report was published in German and hence only accessible to a very limited readership. The English publication allows for high quality review and reception by the international community. 

- Third, the report was very limited in scope. It only presented the meta-analytic summary of the results and no additional analyses (as you could expect from a report that is based on a poster presentation). It was also very limited in theoretical embedding and discussion. The whole report only had four narrow one-column pages.

We have refined the transparency statement to clarify that the German report only contained a summary of the results presented here in full.

We note that you have referenced (Bernstein, D. P., & van den Broek, E. et. al(2009)) which has currently not yet been accepted for publication. Please remove this from your References and amend this to state in the body of your manuscript: (Bernstein, D. P., & van den Broek et al. [Unpublished]”) as detailed online in our guide for authors

Response: Thank you for the advice. We have removed the citation completely.

Reviewer #1

Abstract has not standard way of scribe. In abstract is not 

suitable to use questions. Also is not right to use abstract as a shortened version of paper contain. I propose complete revision of abstract.

Response: We completely rewrote the abstract to better match the requirements and focused on brevity and higher precision in the new version.

The section Introduction is very long but interest. Authors in this section inform reader about research approaches to the different self-states psychotherapeutic and counselling. Becouse the section Introduction is very long, I propose divided this section the section Introduction and section Related work. Section Introduction should be short and concise (contain short information about current state of research problematic and short information what each of the following sections is about).

Response: We apologize for the perceived lack of formal structure in the introduction section. We have now divided this section into several parts as inspired by the reviewer. For further clarity, we have added subheadings. Furthermore, we have added a short organizing paragraph that communicates the purpose and structure of the introduction.

The methods was clar and replicable. I have reservations only to the size of sample. The size 37 persons is very small sample for inferring of research conclusions. 

Response: We apologize for the confusion. The rater sample actually consists of 49 participants and achieved a post-hoc power of 1-β >.99. We have rewritten the section on the sample to clear up any misunderstanding (page 8, line 141). 

As self-assessment authors use Self-Assessment Manikin - that is very suitable method but missing psychologist's opinion and point of view.

Response: Thank you for this comment, we completely agree on the need for clarification. We embedded the description of the method in the psychological literature of emotion theory and assessment (page 10, line 192 – page 11, line 205).

Furthermore, we mention in the generalization/limitations section (limitation 3) that a deeper psychological assessment/rating would be advantageous in future studies (page 20, line 376).

Discussion

I think that this part is the most weak. It should be discussed the relevance and the clinical application, not only in psychotherapy but also in other fields (emotion recognition,...).

Response: Thank you for this great advice. We added paragraphs concerning clinical application and emotion theory to the discussion.

It also the diferent dimensions are not discussed and only in the limitation part are included a discussion about arousal dimension.

Response: Thank you for the helpful advice. We addressed the need for clarification and included a more thorough discussion concerning the different emotional dimensions in the section “Differences between the affective dimensions“ in the discussion (page 18, line 328).

The comment of arousal dimension is not a limitation, is subject of discussion (see above).

Response: We apologize for the lack of structure and have now included this paragraph in the Discussion (page 18, line 333).

Reviewer #2

Reviewer #2: The paper focuses the psychotherapeutic relevance of the self-states, a subject of great clinical relevance.

Ethics statement

I’ve a concern about that authors have not required the ethical approval. Which kind of data protection were adopted?

Response: We appreciate the reviewer’s question and agree with the reviewer’s opinion that ethical considerations are of highest importance. We are happy to explain and clarify our thorough process in the following, parts of which may be specific to Germany where the study was conducted.

It is neither German law nor research standard in Germany to apply for ethics votes for a simple rating study of unproblematical material. If you look at the guidelines of one of the highest research committees in Germany, the RatSWD, you will find the following advice: "RatSWD recommendation (excerpt): “Not all research projects require an ethical assessment by a commission. In many cases, after careful (self-)examination, the project can be classified as unobjectionable in terms of research ethics. This reflection can be promoted significantly and at the same time unbureaucratically through communication with third parties (e.g. peers, supervisors, experts etc.)." (https://www.konsortswd.de/ratswd/best-practice-forschungsethik/forschende/ethikvoten/, translated) 

The rating study of which we present the results here clearly bears no risk for the rating participants, as they view videos without audio and simply rate what they see. Hence, an ethics approval was not necessary. In fact, our IRB doesn’t even give exemption for such studies and only offers full ethical approval procedures as demanded for intervention and medical studies.

However, this shifts the focus to the context of material creation: the coaching sessions. These coaching sessions happened within the teaching context (at Hochschule Zittau/Görlitz) and would have been recorded anyway for documentary (grading) purposes. 

This, in turn, shifts the focus to, firstly, consent and, secondly, data protection.

First, consent for using the video material without audio was given by all filmed clients before the coaching sessions. Second, the data protection steward of Technische Universität Dresden approved our data management and protection concept. This concept included storage of data and removal of audio from the videos for the rating study. Finally, participants of the rating study had to consent to not talk about anything they have seen in the study to third parties.

We hope that this clarifies why we did not seek ethical approval nor could apply for it, but nevertheless strongly cared for ethically sound procedures and for implementing appropriate data protection.

Sample

Have been studies the profile of the students that dropped out early?

Response: We apologize for the lack of information on the dropped out participants and have now included demographic data for the participants in question (page 8, line 145).

“We excluded 37 individuals who dropped out of the experiment early, one who took four hours instead of the intended 1.5 hours, and one who did not change baseline scores on the rating scales in 229 out of the 360 trials. There is no data for 19 of the 37 drop outs since they left the experiment before any data was logged. Among the remaining drop outs 16 were female, 2 were male, and 1 was diverse (Mage = 23.95 years, SDage = 6.52 years).”

Self-Asssessment manikis

A description of the dimensions assessed should be included. Why the authors have not studied the dimension of affect?

Response: Thank you very much for this comment. We actually did study different dimensions of affect. We clarified and extended the according sections in the methods (page 11, line 192), results and discussion (page 18, line 328).

The self-constellation method

Part of this paragraph should be included in the methodology section.

Response: We apologize for the repeated description of the self-constellation method and have removed this paragraph from the discussion and integrated it into the existing description of the self-constellation method in the method section.

Limitations

The authors should explain clearly that for make a generalization more studies should be done... even to reply their results, not only in psychotherapy settings. I think that it could be introduced also in the discussion, not in the limitation section.

Response: Thank you for this advice. We have now included a section named “Generalizability” addressing this point specifically at the end of the general discussion section (page 19, line 347).

In closing, we thank the reviewers for their very helpful and constructive comments and we thank you again for your editorial efforts on this paper. We hope that we addressed sufficiently all of yours and the reviewer’s concerns and are looking forward to your decision.

---

## [Decision Letter · Decision Letter 1]

4 Mar 2024

How self-states help: Observing the embodiment of self-states through nonverbal behavior

PONE-D-23-17809R1

Dear Dr. Scherbaum,

We’re pleased to inform you that your manuscript has been judged scientifically suitable for publication and will be formally accepted for publication once it meets all outstanding technical requirements.

Kind regards,

Giulia Ballarotto

Academic Editor

PLOS ONE

Additional Editor Comments (optional):

Reviewers' comments:

Reviewer's Responses to Questions

**Comments to the Author**

1. If the authors have adequately addressed your comments raised in a previous round of review and you feel that this manuscript is now acceptable for publication, you may indicate that here to bypass the “Comments to the Author” section, enter your conflict of interest statement in the “Confidential to Editor” section, and submit your "Accept" recommendation.

Reviewer #1: (No Response)

Reviewer #2: All comments have been addressed

2. Is the manuscript technically sound, and do the data support the conclusions?

Reviewer #1: Yes

Reviewer #2: Yes

3. Has the statistical analysis been performed appropriately and rigorously? 

Reviewer #1: Yes

Reviewer #2: Yes

4. Have the authors made all data underlying the findings in their manuscript fully available?

Reviewer #1: Yes

Reviewer #2: Yes

5. Is the manuscript presented in an intelligible fashion and written in standard English?

Reviewer #1: Yes

Reviewer #2: Yes

6. Review Comments to the Author

Reviewer #1: I commend the authors for their approach to improving the overall outcome of the submitted research paper. It can be seen that the changes have helped to increase the overall level of research.

Reviewer #2: All concerns have been answered correctly. For me the paper has been improved and it is suitable to publish in your journal.

7. PLOS authors have the option to publish the peer review history of their article (what does this mean?). If published, this will include your full peer review and any attached files.

Reviewer #1: No

Reviewer #2: No

---

## [Editor Report · Acceptance letter]

19 Mar 2024

PONE-D-23-17809R1 

PLOS ONE

Dear Dr. Scherbaum, 

I'm pleased to inform you that your manuscript has been deemed suitable for publication in PLOS ONE. Congratulations! Your manuscript is now being handed over to our production team.

Kind regards, 

on behalf of

Dr Giulia Ballarotto 

Academic Editor

PLOS ONE